

# 1  Bed topography of Princess Elizabeth Land in East Antarctica

**Xiangbin Cui[1], Hafeez Jeofry[2,3], Jamin S Greenbaum[4], Jingxue Guo[1], Lin Li[1], Laura E Lindzey[5],**
**Feras A Habbal[6], Wei Wei[4], Duncan A Young[4], Neil Ross[7], Mathieu Morlighem[8], Lenneke M.**
**Jong[9,10], Jason L Roberts[9,10], Donald D Blankenship[4], Sun Bo[1] and Martin J. Siegert[11]**
[1] Polar Research Institute of China, Jinqiao Road, Shanghai, China
[2] Faculty of Science and Marine Environment, Universiti Malaysia Terengganu, Kuala Terengganu, Terengganu, Malaysia
[3] Institute of Oceanography and Environment, Universiti Malaysia Terengganu, Kuala Terengganu, Terengganu, Malaysia
[4] Institute for Geophysics, Jackson School of Geosciences, The University of Texas at Austin, Austin, Texas, USA
[5] Department of Ocean Engineering, Applied Physics Laboratory, University of Washington, USA
[6] Oden Institute for Computational Engineering and Sciences, University of Texas at Austin
[7] School of Geography, Politics and Sociology, Newcastle University, Newcastle upon Tyne, UK
[8] Department of Earth System Science, University of California Irvine, Irvine, California, USA
[9] Australian Antarctic Division, Kingston, Tasmania, Australia
[10] Institute for Marine and Antarctic Studies, University of Tasmania, Hobart, Tasmania
[11] Grantham Institute and Department of Earth Science and Engineering, Imperial College London, South Kensington,
London, UK

## 20  Abstract

We present a topographic digital elevation model (DEM) for Princess Elizabeth Land (PEL), East
Antarctica – the last remaining region in Antarctica to be surveyed by airborne radio-echo sounding
(RES) techniques. The DEM covers an area of ~900,000 km$^2$ and was established from new RES data
collected by the ICECAP-2 consortium, led by the Polar Research Institute of China, from four
campaigns since 2015. Previously, the region (along with Recovery basin elsewhere in East Antarctica)
was characterised by an inversion using low resolution satellite gravity data across a large (>200 km
wide) data-free zone to generate the Bedmap2 topographic product. We use the mass conservation
(MC) method to produce an ice thickness grid across faster-flowing (>30 m yr$^{-1}$) regions of the ice sheet
and streamline diffusion in slower-flowing areas. The resulting ice thickness model is integrated with
an ice surface model to build the bed DEM. With the revised bed DEM, we are able to model the flow
of subglacial water and assess where the hydraulic pressure, and hydrological routing, is most sensitive
to small ice-surface gradient changes. Together with BedMachine Antarctica, and Bedmap2, this new
PEL bed DEM completes the first order measurement of subglacial continental Antarctica – an
international mission that began around 70 years ago. The ice thickness and bed elevation DEMs of
PEL (resolved horizontally at 500 m relative to ice surface elevations obtained from a combination of
European Remote Sensing Satellite 1 radar (ERS-1) and Ice, Cloud and Land Elevation Satellite (ICESat)
laser satellite altimetry datasets) are accessible from https://doi.org/10.5281/zenodo.3666088 (Cui et
al., 2020).

## 40  1.  Introduction


Radio-echo sounding (RES) is commonly used to measure ice thickness, and to understand subglacial
topography and basal ice-sheet conditions (Dowdeswell and Evans, 2004; Bingham and Siegert, 2007).
A series of airborne geophysical explorations were conducted across East Antarctica in the 1970s by
the Scott Polar Research Institute (SPRI) (Robin et al., 1977; Dean et al., 2008; Turchetti et al., 2008;



Naylor et al., 2008), which led to the first compilation 'folio' maps of subglacial bed topography, ice-
sheet surface elevation and ice thickness of Antarctica (Drewry and Meldrum, 1978; Drewry et al.,
1980; Jankowski and Drewry, 1981; Drewry, 1983). Since then, multiple efforts have been made to
collect and compile RES data in order to expand the RES database across the continent (Lythe et al.,
2001; Fretwell et al. 2013). Russian glaciologists conducted the first geophysical exploration of the
coast of Princess Elizabeth Land (PEL) between 1971–2016, providing basic ice thickness, bed
topography and magnetic field data (Popov and Kiselev, 2018; Popov, 2020). To date, virtually no RES
data have been acquired upstream of ~300 km from the grounding line of PEL. Hence, this region has
been described as one of the so-called 'poles of ignorance' (Fretwell et al., 2013) and its representation
in recent bed DEMs is as a zone of flat topography, reflecting the absence of RES data (Morlighem et
al., 2020). Indeed other data gaps (Recovery system, Diez et al., 2019; and South Pole, Jordan et al,
2018) have been filled recently, leaving PEL has the last remaining site to be surveyed systematically.
In the absence of bed data, glaciologists have had to rely on satellite imagery, inversion from poor
resolution satellite gravity observations, and ice-flow modelling to infer the subglacial landscape and
its interaction with the ice above (Fretwell et al., 2013; Jamieson et al., 2016). For example,
combination of three satellite-derived mosaics, and some initial exploratory RES data, have been used
to hypothesise the subglacial features of PEL (Jamieson et al., 2016). That study utilised the first RES
data collected as part of the collaborative effort between the US–UK–Australian ICECAP (International
Collaborative Exploration of Central East Antarctica through Airborne geophysical Profiling), which
was conducted between 5$^{th}$ December 2010 to 20$^{th}$ January 2013 (Blankenship et al., 2017). Jamieson
et al. (2016) reveal presence of a potentially large (>100 km long) subglacial lake (white box; Figure 1)
and an expected canyon morphology across the PEL sector (Jamieson et al., 2016). Previously, a study
by Dongchen et al. (2004) adopted the interferometric synthetic-aperture radar (InSAR) satellite
technology to generate an 'experimental' subglacial bed elevation model across the Grove Mountains
(Figure 1a). While the result contains a level of 'detail', it has an obvious limitation in that the bed
elevation was based solely on the satellite data and without direct measurement of the subglacial
landscape. Another study used an inversion technique to generate a 'synthetic' glacier thickness of
the PEL region from satellite gravity data, as part of the Bedmap2 compilation (Fretwell et al., 2013).
A qualitative inspection of the Bedmap2 bed elevation product reveals the bed of PEL to be
anomalously flat –a consequence of its use of satellite gravity data in a low resolution inversion for
bed elevation across a data-free region. Hence, the bed topography in PEL is the poorest-defined of
any region in Antarctica – and indeed of any land surface on Earth.
Here, we present the first detailed ice thickness DEM for PEL, based on new RES measurements
collected by the ICECAP2 programme led by the Polar Research Institute of China (PRIC) since 2015.
We integrated the DEM with ice surface elevation measurements to produce a bed DEM. We briefly
discuss the differences between the new bed DEM and its representation in Bedmap2, and the impact
of the new DEM on calculations of the flow of subglacial water. The bed DEM is relative to ice surface
elevations from a combination of European Remote Sensing Satellite 1 radar (ERS-1) and Ice, Cloud
and Land Elevation Satellite (ICESat) laser satellite altimetry datasets (Bamber et al., 2009). The ice
thickness DEM can be easily integrated with updated surface DEMs in future (Helm et al., 2014; Howat
et al., 2019) and, in particular, the upcoming Bedmap3 product.



## 2. Study Area

The PEL sector of East Antarctica is bounded on the west by the Amery Ice Shelf, and on the east by Wilhelm II Land (Figure 1a). The region covered by the new DEM we present here extends ~1,300 km from East to West and ~800 km from North to South. In comparison with Bedmap2, the new DEM benefits from recently acquired airborne geophysical data collected by the ICECAP2 programme over four austral summer seasons from 2015 to 2019 (Figure 1b). We use the Differential Interferometry Synthetic Aperture Radar (DInSAR) grounding line (Rignot et al., 2011) to delimit the ice-shelf facing margin of the ice sheet.

## 3. Data and Methods

During the first ICECAP2 season (2015/16), a survey acquiring exploratory 'fan-shaped' radial profiles, to maximize range and data return on each flight, was completed across the broadly unknown region of PEL. These flight lines extend from the coastal Progress Station to the interior ice-sheet divide at Ridge B (Figure 1a). In the second and third seasons (2016/17 and 2017/18), a survey 'grid' was completed, targeting enhanced resolution over a proposed subglacial lake and a series of basal canyons (see Jamieson et al., 2016). In the fourth season (2018/19), a few additional transects were completed to fill the largest data gaps within aircraft range.

Field data acquisition was achieved using the "Snow Eagle 601" aerogeophysical platform; a BT-67 airplane operated by the Polar Research Institute of China for the Chinese National Antarctic Research Expedition (CHINARE) program (Figure 2a and b). The suite of instruments configured on the airplane include a phase coherent RES system, functionally similar to the High Capability Airborne Radar Sounder developed by the University of Texas Institute for Geophysics (UTIG), which has been used on many ICECAP surveys (i.e. Young et al., 2011; Greenbaum et al., 2015). HiCARS is a phase coherent RES system, operating at a central frequency of 60 MHz and a peak power of 8 kW, making it capable of penetrating deep (>3 km) ice in Antarctica. After applying coherent integration and pulse compression at a bandwidth of 15 MHz, which gave an along-track spatial sampling rate and a vertical resolution of ~20 m and ~5.6 m, respectively. Further details on the parameters and introduction of the CHINARE IPR can be found in Cui et al. (2018). A JAVAD GPS receiver and its four antennas are mounted at the aircraft centre of gravity (CG), tail and both wings. GPS data from antenna at the aircraft CG were used for RES data interpretation.

## 4. Data Processing

Ice thickness measurements were derived from two RES data products from which the ice-bed interface was traced and digitized: (a) 2D focused SAR processed data applied to RES data from the first two seasons; and (b) unfocused 'field' RES data from the third and fourth seasons. Raw RES data were first separated to differentiate PST (Project/Set/Transect) during the field data processing. Pulse compression, filtering, 10-traces coherent stacking and 5-traces incoherent stacking were then applied to generate a field RES data product. The field RES data can be used for quality control and is also good enough for initial ice-bed interface measurements, from which a first-order ice thicknesses and bed elevation DEM can be calculated. To achieve better-quality RES images, two-dimensional focused SAR processing was applied to data from the first two seasons (Peters et al., 2007). The ice-bed interface



was picked in a semi-automatic manner using a picking program used previously by the ICECAP
program on data from the Aurora and Wilkes subglacial basins (Blankenship et al., 2016; Blankenship
et al., 2017). Ice thicknesses are calculated from multiplying two-way travel time by the velocity of
electromagnetic waves in ice (i.e. 0.168 m ns$^{-1}$) (Cui et al., 2018). Firn corrections were not applied.
The precise point positioning (PPP) method was used in the GPS processing to improve positioning
accuracy since the flight distance is too far from the GPS base station for post airborne GPS data
processing. Processed GPS data are interpolated and fitted to the radar traces according to time
stamps generated by the integrated airborne system. Aircraft to ice-surface range was calculated by
multiplying the two-way travel time of the radar reflections of the ice surface by its velocity in air (0.3
m ns$^{-1}$). Figure 2c shows examples of the two-ways RES images from the data collected in 2017/18.

*4.1 Quantifying ice thickness, bed topography and subglacial hydrology pathway*
To derive the ice thickness map based on the PEL radar measurements, we employed a variety of
techniques depending on the ice speed following the approach described in Morlighem et al. (2020).
In fast flowing regions (i.e. velocity >30 m yr$^{-1}$), we relied on mass conservation (MC; Figure 3),
constrained by the PEL RES data and additional RES data that were available as part of BedMachine
Antarctica (Morlighem et al., 2020). In the slower moving regions inland, we relied on a streamline
diffusion interpolation to fill between data points (Figure 3).

150        For the purpose of comparing the bed DEM with Bedmap2, the 1 km ice-surface elevation DEM
from Bamber et al. (2009) was used. Prior to the subtraction process, the ice thickness was resampled
using the 'Nearest Neighbour' function in ArcGIS to a 1 km spacing and referenced to the polar
stereographic projection (Snyder, 1987). The PEL ice thickness model was then subtracted from the
Bamber et al. (2009) ice surface elevation DEM to produce a 1 km bed DEM (Figure 4b). The Bedmap2
bed DEM was transformed from the g104c geoid vertical reference to the WGS 1984 vertical reference
frame (Figure 4c). A difference map was then computed by subtracting the Bedmap2 bed DEM from
the ICECAP2 bed DEM (Figure 4d). Crossover analyses show RMS errors of 24.2 m (2015/16), 39.2 m
(2016/17), 10.4 m (2017/18), 7.5 m (2018/19) and 35.4 m (for the full dataset).

159        Modelling subglacial water flow for both DEMs utilized the ice-surface elevation from a
combination of European Remote Sensing Satellite 1 radar (ERS-1) and Ice, Cloud and Land Elevation
Satellite (ICESat) laser satellite altimetry datasets (Bamber et al., 2009). Subglacial hydrology pathways
for both the Bedmap2 and our bed DEMs (Figure 4e) were determined by assuming the pressure
equilibrium between ice overburden and basal water (Shreve, 1972) represented by the following
equation:

$\varphi = g(\rho_w y + \rho_i h)$         (1)

where $\varphi$ is the theoretical hydropotential surface (Figures 4f and 4g), $y$ is the bed elevation, $h$ is the
ice thickness, $\rho_w$ and $\rho_i$ are the density of water (1000 kg m$^{-3}$) and ice (920 kg m$^{-3}$) respectively,
assuming ice to be homogenous, and $g$ is the acceleration due to gravity (9.81 ms$^{-2}$). Hydrological
sinks were filled in the hydropotential surface to produce realistic hydrology pathways, and flow
direction was applied by assigning a direction from eight adjacent cells (i.e. D8 approach) to determine
the steepest downslope neighbouring cell (Jenson and Domingue, 1988).





## 5. Results

*5.1 Subglacial morphology of Princess Elizabeth Land*

The new RES data allow us to form an appreciation of the subglacial topography of PEL (Figure 4a and b). While its hypsometry (Figure 5) reveals an area-elevation distribution that is mainly concentrated around 0 to 500 m (>15% frequency, Figure 5a) with a mean elevation of 347.29 m, the DEM reveals a newly-discovered broad, low-lying subglacial basin (>400 m below sea level) (Figure 4b). This is the most distinct new topographic feature uncovered by the new data. The data also resolve higher ground across the northwest grid of the PEL DEM (i.e. American Highland, Figure 5a). A deep (i.e. ~500 m below sea level) subglacial trough can be observed near to Zhaojun Di area, coinciding with the location of fast ice flow towards the Amery Ice Shelf (Figure 1a). Mountains beneath Ridge B (Figure 1a) can be observed in enhanced resolution from the new data (Figure 5b) with an average elevation of ~1500 m above sea level. The bed topography closer to the grounding line (i.e. Wilhelm II Land) and at the central grid areas are characterized as having a lower bed elevation (below sea level, Figure 5b), consistent with the recent BedMachine Antarctica product (Morlighem et al., 2020). Subglacial troughs with depth less than ~500 m can also be observed in Wilhelm II Land.

*5.2 Comparison with Bedmap2*

The 1 km bed elevation model of the PEL sector of East Antarctica, the corresponding Bedmap2 DEM and a map displaying difference between the two are shown in Figure 4b,c,d. The new DEM reveals substantial changes relative to Bedmap2 bed product especially across the central upstream region of PEL. For example, the PRIC bed DEM shows noticeable disagreement from Bedmap2 across the Australian Antarctic Territory extending from the central grid of the DEM (i.e. Korotkevicha Plateau and King Leopold and Queen Astrid Coast) to the Mason Peaks at the northern grid, with differences typically ranging between -100 and -300 m. However, the bed elevation is higher in the new bed DEM compared with Bedmap2 across Wilhelm II Land with a mean of ~90 m. Because the new bed DEM is higher in some places compared with Bedmap2, and lower in others, the mean difference for the entire PEL study area is only -41 m.

We also present five terrain profiles for both DEMs (Figure 6), which collectively cover most of the PEL sector (Figure 1b). The purpose is to capture as much of the subglacial morphology as possible and assess the accuracy of the DEMs in their characterization of these subglacial features. In general, and as one would expect, the ICECAP2 bed DEM shows reasonable agreement with the RES transects in all profiles compared with Bedmap2 bed DEM. Consistencies between the ICECAP2 DEM and the bed elevation from RES data picks can be seen upstream of the PEL DEM grid (i.e. Mason Peaks and Zhaojun Di) with a correlation coefficient of 0.54 (RE:15%) and 0.94 (RE:16%) for Profile A and B, respectively. This is higher relative to Bedmap2 DEM which is 0.45 (RE:25%) for Profile A and 0.86 (RE:24%) for Profile B. A significant improvement is noted in the new DEM with correlation coefficient of 0.79 (RE:21%), compared with 0.50 (RE:30%) for Bedmap2, across the American Highland in Profile C (Figure 6). A slightly lower correlation coefficient than Profile C is quantified for the new DEM, at 0.78 (RE:38%), but it is still higher than in Bedmap2 at 0.60 (RE:42%) for Profile D. Similarly, the correlation coefficient between both DEMs near to the Wilhelm II Land (Figure 6; Profile E–E') is higher in the new DEM at 0.91 (RE:34%) than Bedmap2 at 0.57 (RE:42%).



*5.3 Subglacial hydrology and lakes*

Understanding subglacial water flow in Antarctica is crucial for assessing its potential influence on ice-sheet flow and dynamics (Stearns et al., 2008). The main flow network for most parts remains broadly unchanged irrespective of the bed DEM used, highlighting the dominance of ice surface slopes on defining subglacial water pathways (Wright et al., 2008; Le Brocq et al., 2009; Horgan et al., 2013). Nevertheless, there are a few clear differences in the subglacial hydrological pathways, particularly across the central region of the grid where the bed differences are greatest.

Essentially, there are five main subglacial water networks observed, where the hydrological pathway is related to geomorphology (Figure 4e). One subglacial water pathway (red region, Figure 4e), can be seen feeding into a subglacial trough (Figure 4b, 4c and 5b), coinciding with the location of fast ice flow (Figure 1a). A number of subglacial lakes exist at the southwest grid of the DEM close to Lake Vostok, the largest subglacial lake in East Antarctica first detected in 1974 (Figure 1a; Robin et al., 1977; Kapitsa et al., 1996; Studinger et al., 2004; Siegert et al., 2011; Siegert, 2017), which may provide water to this network. It should be noted that Lake Vostok lies just outside the PEL DEM. There are seven previously recognised subglacial lakes located at the southwestern region of the DEM which are named Sovetskaya, 90°E, SPRI-47, SPRI-54/59, SPRI-60, C25SAE1 and C25SAE2 (Figure 1a; Wright and Siegert, 2012), indicating the presence of stored water beneath the ice-sheet interior of PEL. In addition, subglacial lakes named Komsomolskoe and R15Ea_4 are located at the southern grid of the DEM (Figure 1a; Wright and Siegert, 2012).

Another subglacial water network (yellow region, Figure 4e) can be seen beside the American Highlands, where a newly discovered subglacial lake has been proposed. A study by Jamieson et al. (2016) suggested an extensive and elongated smooth-surface feature (white box; Figure 1), elucidated as a subglacial lake based on the similar characteristics of the ice surface with subglacial lakes identified previously using the same satellite dataset (Bell et al., 2006; Bell et al., 2007). The feature covered an area of ~ 1250 km$^2$, which would be the second largest by length after Lake Vostok (Kapitsa et al., 1996) and the fourth largest by area in Antarctica after Lakes Vostok, 90°E and Sovetskaya (Bell et al., 2006). The ice thickness above the subglacial lake is shown to be ~600 m greater in the PEL DEM relative to Bedmap2 (Figure 4b, 4c and 4d), but even this may lead to an under appreciation of the real bed topography owing to the unknown thickness of the water layer. A series of shorter and distinct subglacial water pathways (orange region, Figure 4e) can be seen flowing towards the grounding line (i.e. Progress Station and Ranvik Glacier). Two separated subglacial water networks (blue and green regions, Figure 4e) are observed across the Wilhelm II Land, coinciding with the location of subglacial troughs (Figure 4b, 4c and 5b) and the ice surface features noted from satellite images (Jamieson et al., 2016).

## 6. Data availability

The ICECAP2 ice thickness and bed elevation models of the PEL sector are available in 500 m horizontal resolutions at https://doi.org/10.5281/zenodo.3666088 (Cui et al., 2020). The airborne radio-echo sounder ice thickness measurements used to generate the products, recorded here in comma-separated values (CSV) format is accessible from https://doi.org/10.5281/zenodo.3815064. The 1 km



ice-sheet surface elevation DEM derived using a combination of ERS-1 surface radar and ICESat laser
altimetry is downloadable from the National Snow and Ice Data Center (NSIDC) website at
https://nsidc.org/data/docs/daac/nsidc0422_antarctic_1km_dem/. If the users wish to modify the
bed DEM, our model can be easily integrated with the updated surface elevation models (Helm et al.,
2014; Howat et al., 2019). Auxiliary details for the MEaSUREs InSAR ice velocity map of Antarctica can
be found at https://doi:10.5067/MEASURES/CRYOSPHERE/nsidc-0484.001. The satellite images for
MODIS Mosaic of Antarctica 2008-2009 and RADARSAT (25m) are obtainable from
https://doi.org/10.7265/N5KP8037      and      https://research.bpcrc.osu.edu/rsl/radarsat/data/,
respectively. A summary of the data used in this paper and their availability is provided in the Table 1.
**7.  Summary**

We have compiled the first airborne RES dataset for PEL; acquired by ICECAP2 and led by PRIC. From
the data, using a combination of interpolation and modelling techniques, we have generated a bed
DEM which is gridded to 1 km resolution for direct comparison with Bedmap2, and at a higher
resolution of 500 m for ice sheet modelling. The DEM has a total area of ~899,730 km$^2$. Considerable
variabilities between the new DEM and Bedmap2 are observed, particularly at the central grid of the
DEM where a broad subglacial basin occurs, and across the Wilhelm II Land toward the margin. The
PEL DEM completes the first-order data coverage of subglacial Antarctica – a feat spanning around 70
years of international collaboration.


**Acknowledgements**
This research was supported by the Chinese Polar Environmental Comprehensive Investigation and
Assessment Programs (CHINARE-02-02), the National Natural Science Foundation of China
(41941006) and the National Key R&D Program of China (2019YFC1509102). MJS acknowledges
support from the British Council's Global Innovation Initiative between the UK, USA, China and India.
We thank the volunteers at QGIS for open-source software used to draw many of the figures in this
paper. DDB, JG and DY acknowledge the G. Unger Vetlesen Foundation, and US National Science
Foundation grants PLR-1543452 and PLR- 1443690. JR acknowledges the Australian Antarctic
Division, which provided funding and logistical support (AAS 4346 and 4511). This work was also
supported by the Australian Government's Cooperative Research Centres Programme through the
Antarctic Climate & Ecosystems Cooperative Research Centre and under the Australian Research
Council's Special Research Initiative for Antarctic Gateway Partnership (Project ID SR140300001).
This is UTIG contribution ####.

**Competing Interests**
The authors report no competing interests for this paper.

**Author contributions**
This paper was research and written by the ICECAP2 partnership, in which all authors are members.
Specific responsibilities are as follows. XB, JSG, JG, LL, LEL, FH, WW, LJ and JRL undertook fieldwork
and data acquisition. JSG and DAY undertook data processing. MM and HJ undertook data
interpolation. All authors comments and edited drafts of this paper. The paper was written by MJS
and HJ. ICECAP2 is led by SB, JLR, DDB and MJS.



**Table 1:** Data files and locations.

| Products | Files | Location | DOI/URL |
|---|---|---|---|
| Bed elevation DEM | 500m bed elevation DEM | Zenodo Data Repository Cui et al. (2020) | https://doi.org/10.5281/zenodo.3666088 |
| Ice thickness DEM | 500m ice thickness DEM | Zenodo Data Repository Cui et al. (2020) | https://doi.org/10.5281/zenodo.3666088 |
| Airborne ice thickness data | Polar Research Institute of China ice thickness data in CSV format | Zenodo Data Repository Cui et al., (2020) | https://doi.org/10.5281/zenodo.3815064 |
| 1 km ice sheet surface DEM | ERS-1 radar and ICESat laser satellite altimetry | National Snow and Ice Data Center (NSIDC) | https://nsidc.org/data/docs/daac/nsidc0422_antarctic_1km_dem/ |
| Ice velocity map of Central Antarctica | MEaSUREs InSAR-based ice velocity | National Snow and Ice Data Center (NSIDC) | https://doi:10.5067/MEASURES/CRYOSPHERE/nsidc-0484.001 |
| Ice sheet surface satellite imagery | MODIS Mosaic of Antarctica (2008 – 2009) (MOA2009) | National Snow and Ice Data Center (NSIDC) | https://doi.org/10.7265/N5KP8037 |
| | RADARSAT (25m) satellite imagery | Byrd Polar and Climate Research Center | https://research.bpcrc.osu.edu/rsl/radarsat/data/ |




(a)

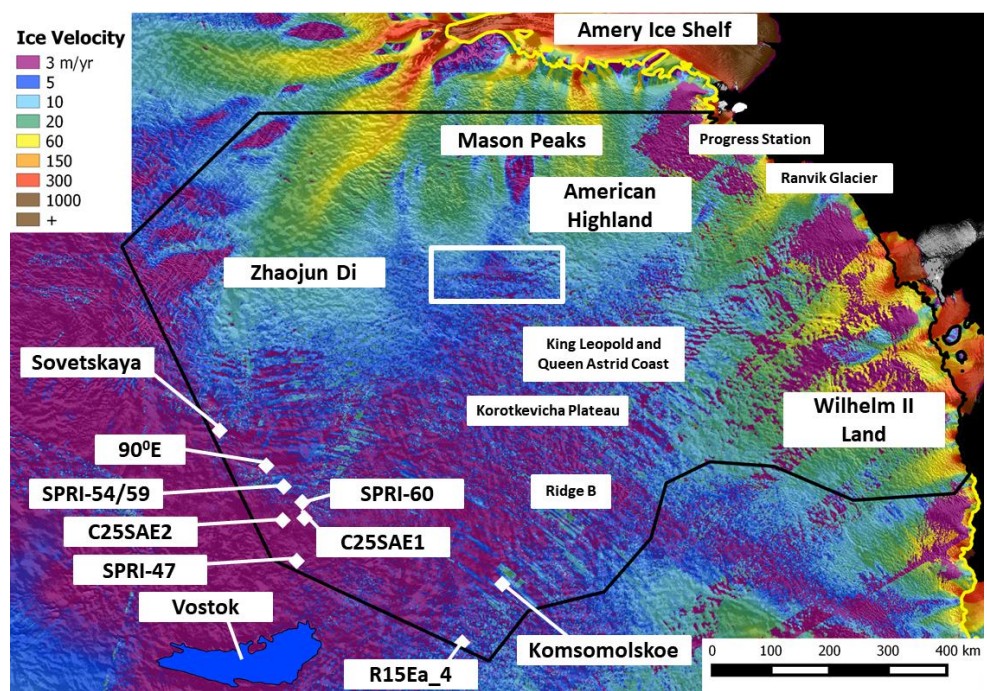

(b)

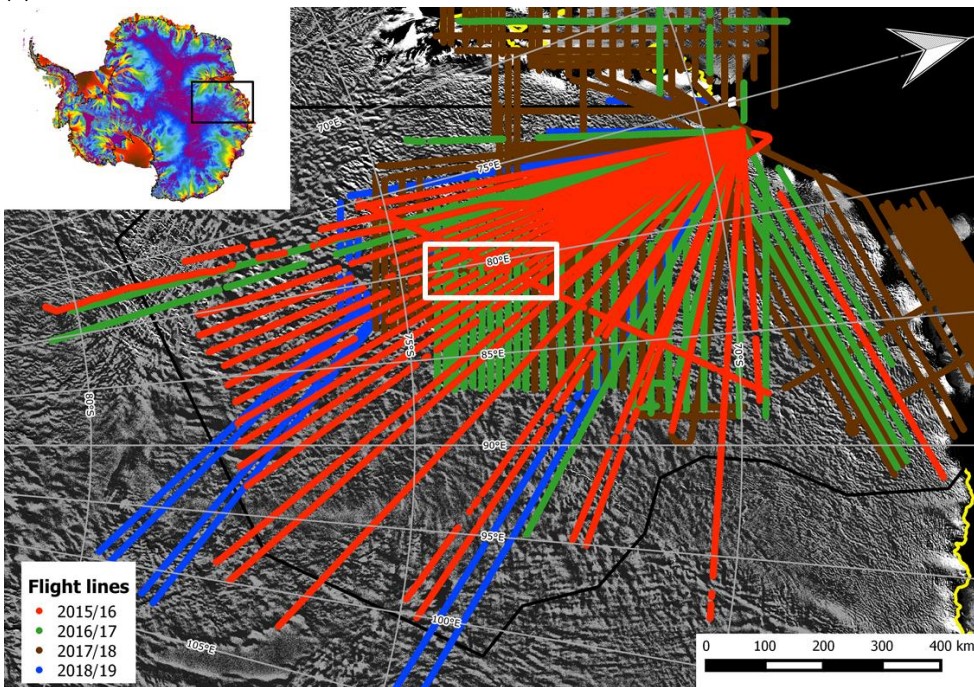




(c)

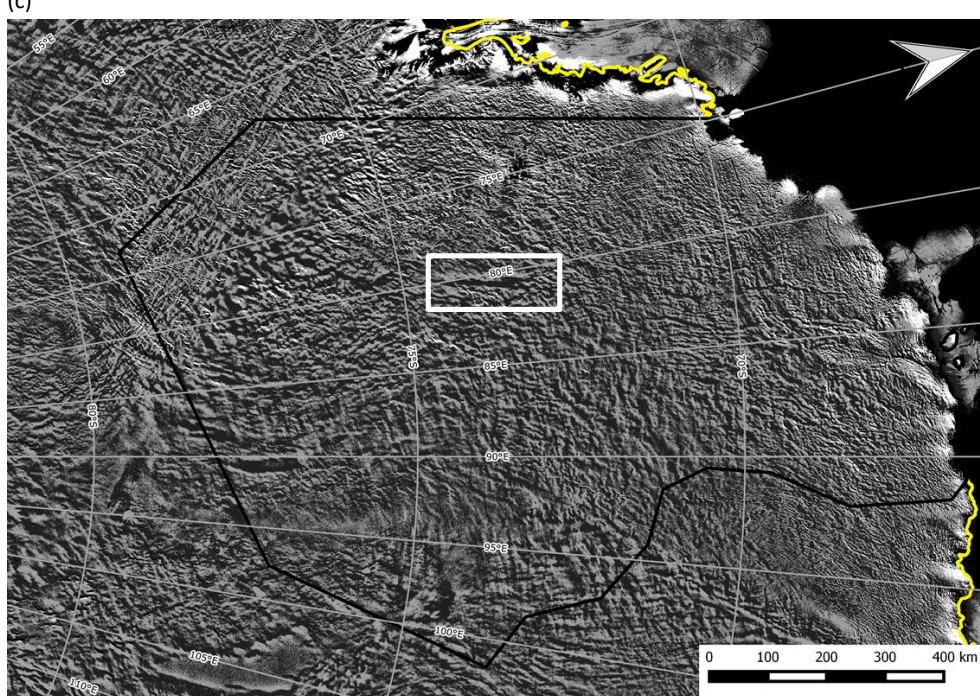

**Figure 1.** Map of (a) ice flow velocity version 2 (Rignot et al., 2017b); (b) the Aerogeophysical flight
lines surveyed by PRIC in four seasons which are 2015/16 (orange), 2016/17 (green), 2017/18 (red)
and 2018/19 (blue) across the PEL sector; the inset denotes location of the study region in East
Antarctica. Both images are overlain by MODIS Mosaic of Antarctica 2008–2009 (Haran et al., 2014);
and (c) MODIS Mosaic of Antarctica 2008–2009 satellite image (Haran et al., 2014). The black line
denotes the grid boundary for PEL bed elevation model. White box indicates a location of a previously
discovered smooth-surface elongated and extensive feature interpreted as a potential subglacial lake
(Jamieson et al., 2016). The Differential Interferometry Synthetic Aperture Radar (DInSAR) grounding
line (yellow line) are also shown (Rignot et al., 2017a).

(a)

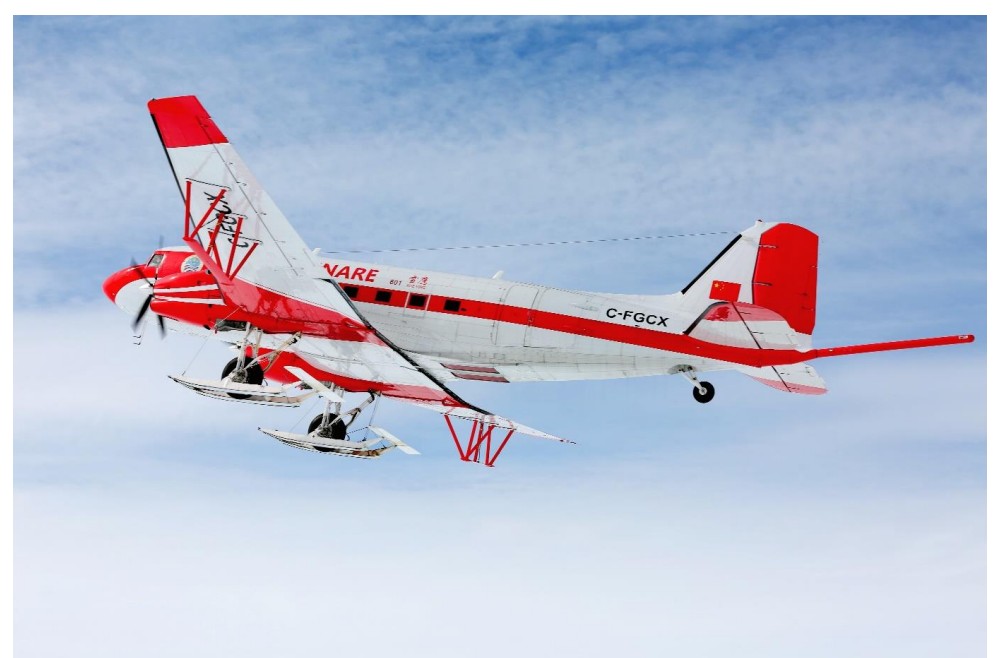


(b)

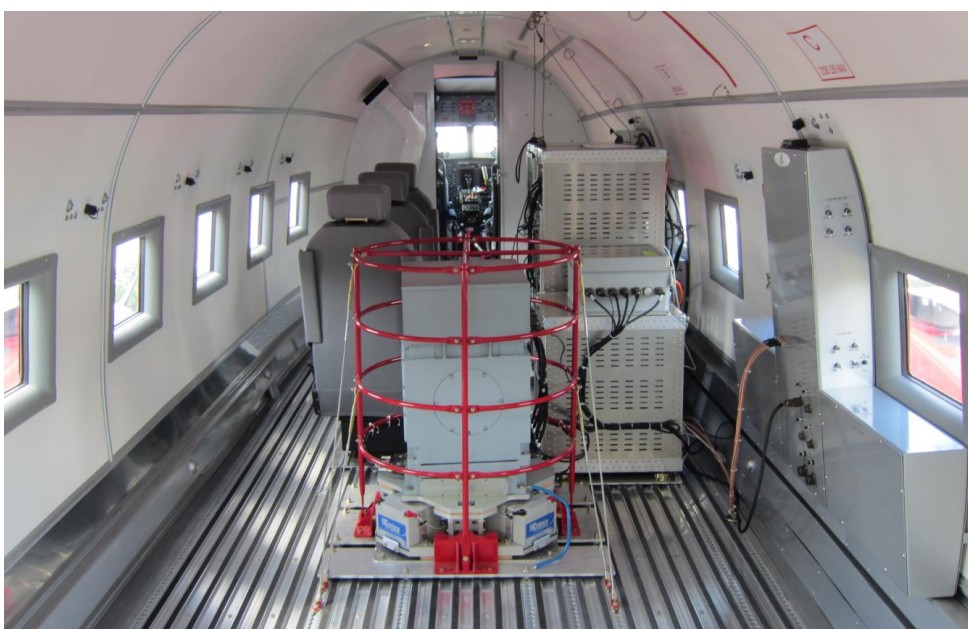




(c)

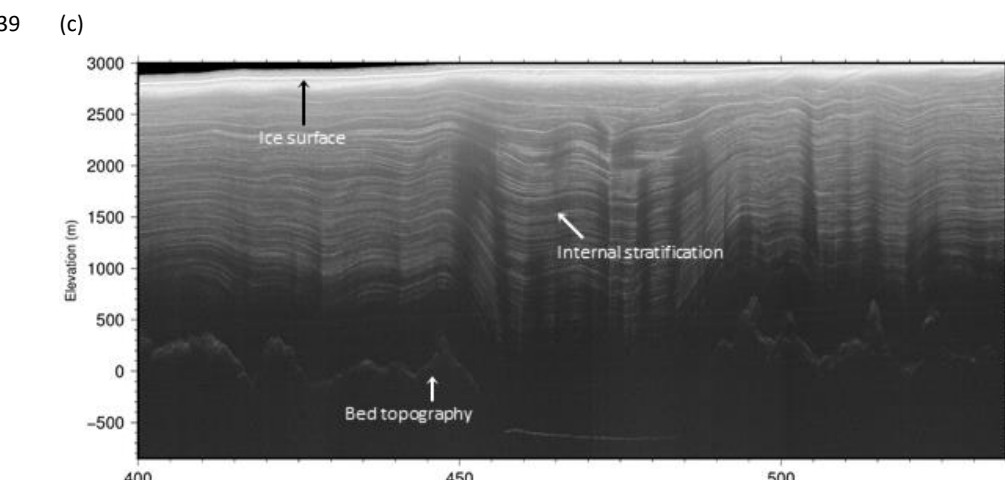


**Figure 2.** (a) Snow Eagle 601 airplane operated by the Polar Research Institute of China for the Chinese
National Antarctic Research Expedition (CHINARE) program; (b) The interior image of the airplane
showing the airborne radio-echo sounder equipment; and (c) Two-dimensional radio-echo sounding
radargram collected in 2017/18 revealing the quality of internal layers, bed topography and subglacial
lake water.




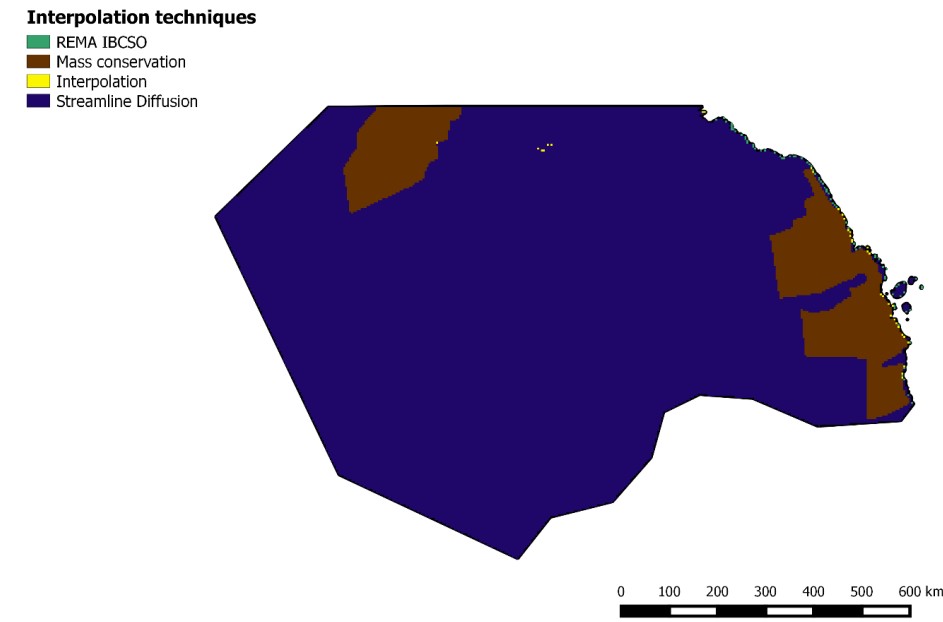


**Figure 3.** Map shows interpolation techniques used to infer ice thickness DEM across PEL, reference
Elevation Model of Antarctica, International Bathymetric Chart of the Southern Ocean (REMA IBCSO,
green), mass conservation (brown), interpolation (yellow) and streamline diffusion (blue).





(a)

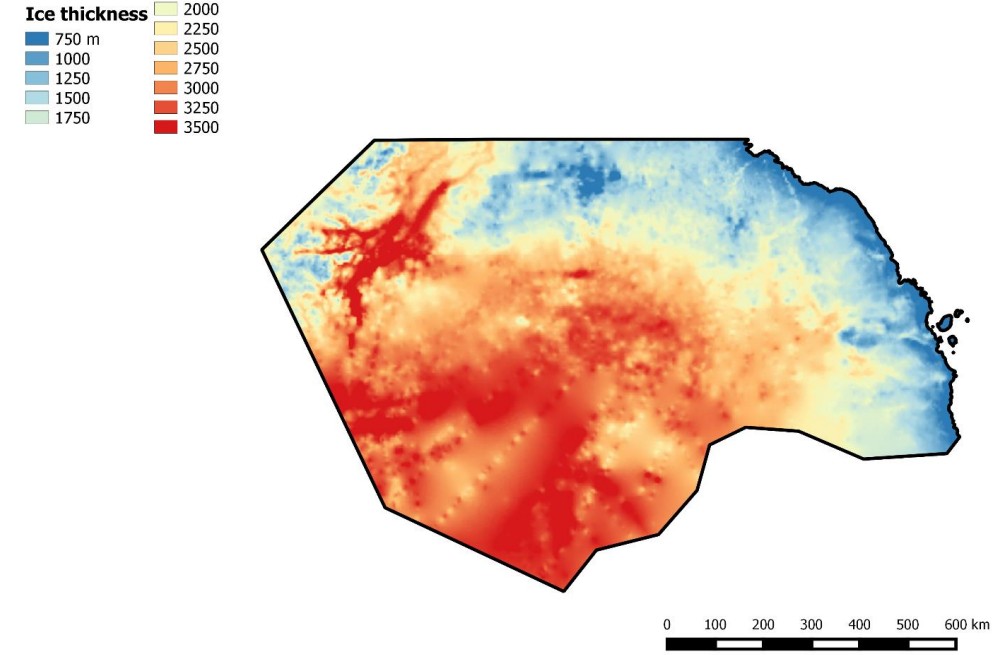

(b)

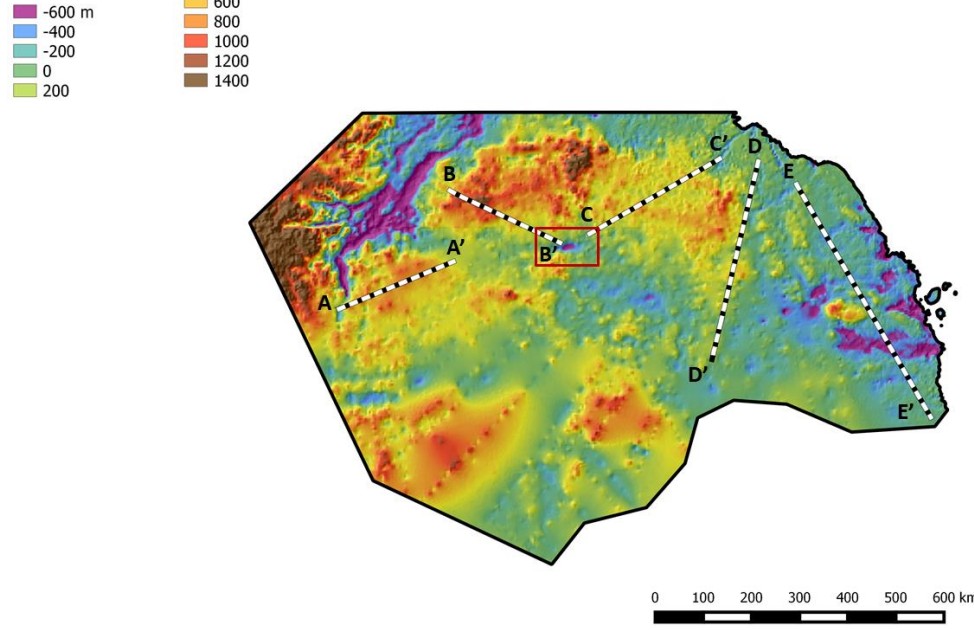




(c)

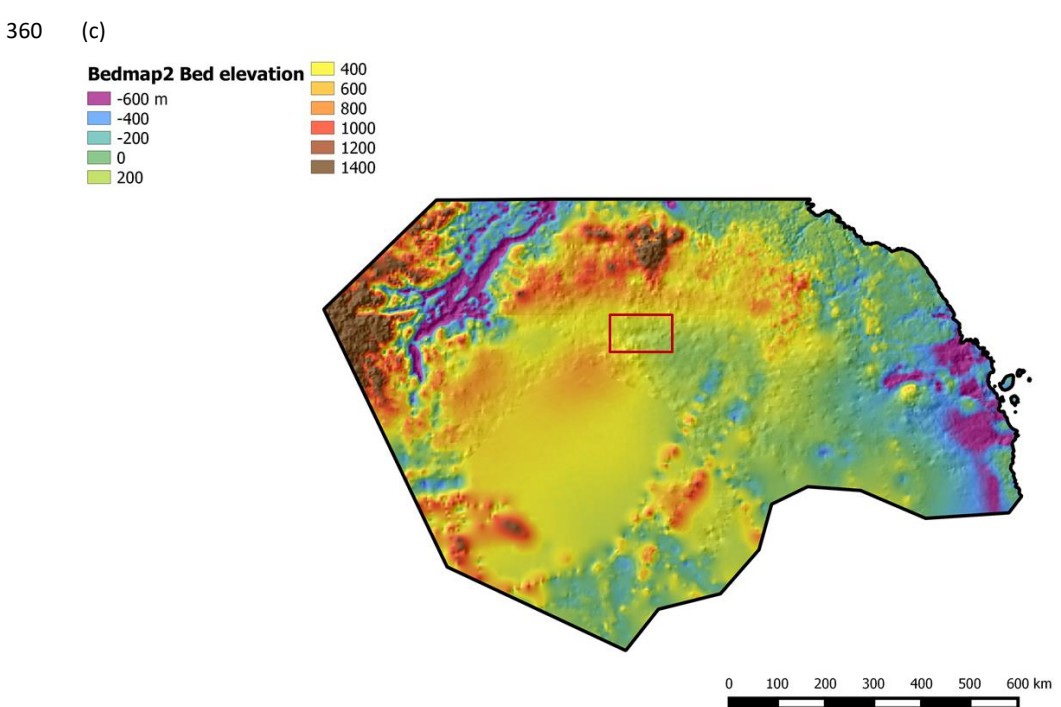

(d)

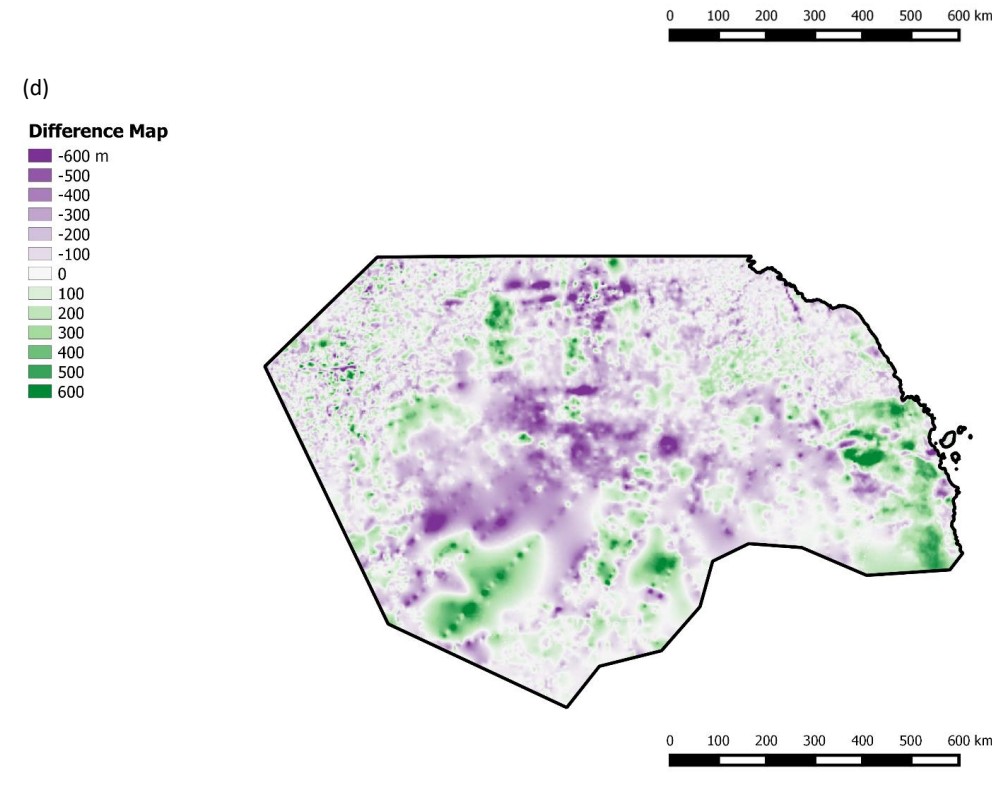




(e)

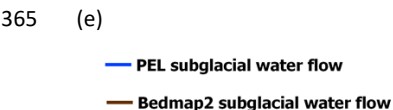

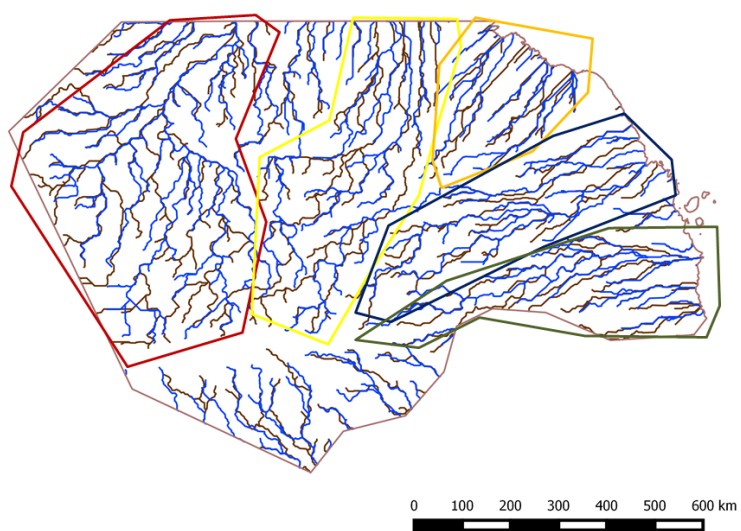

(f)

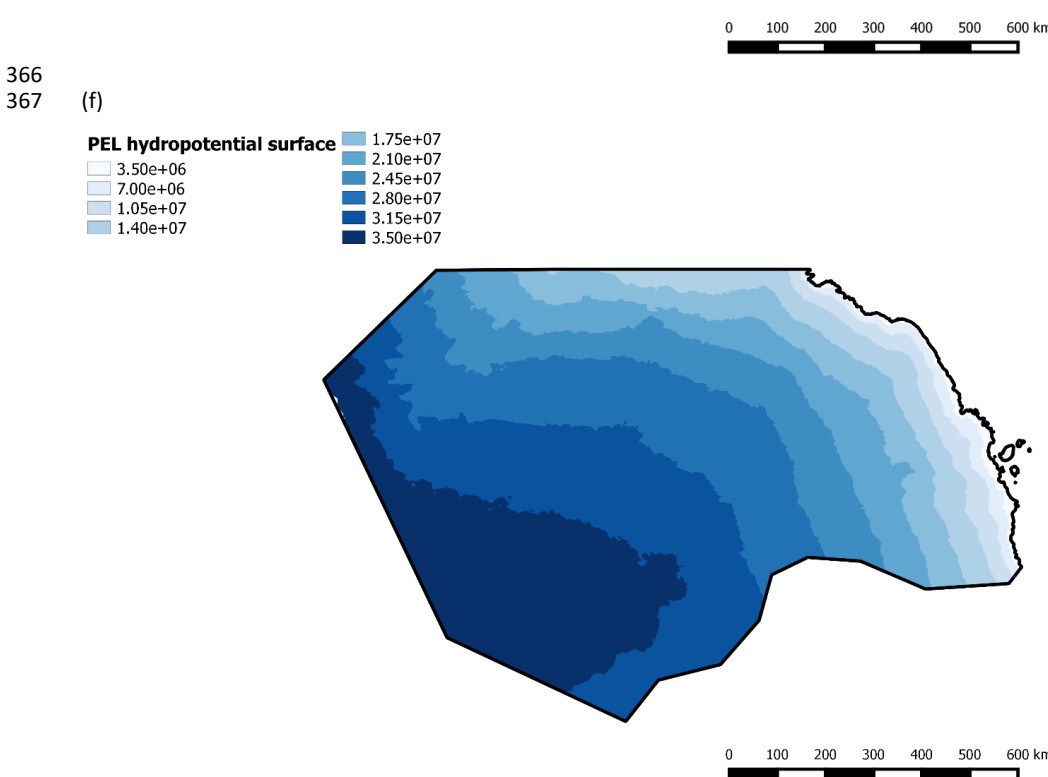


(g)

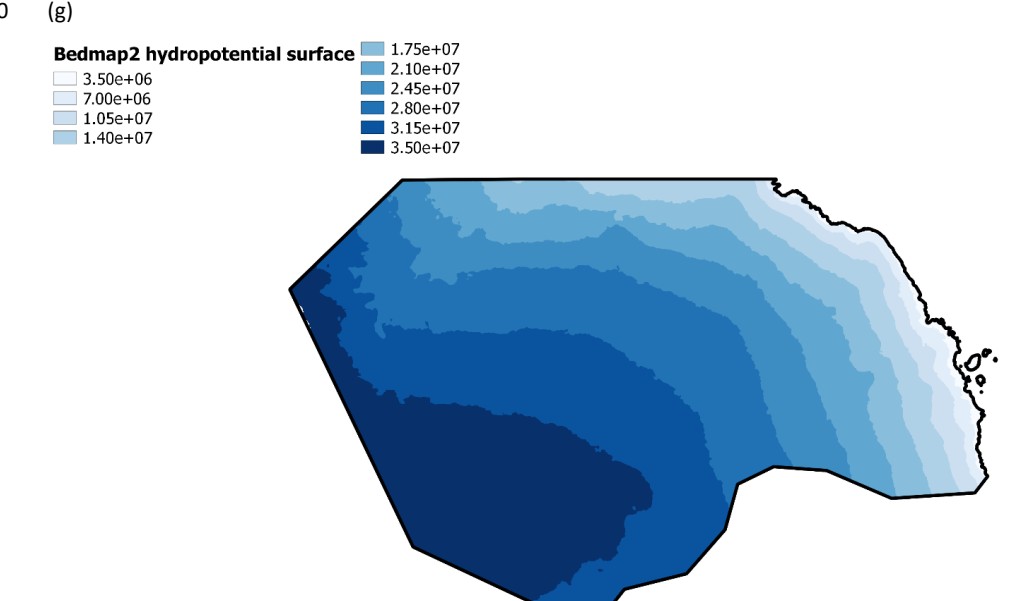



**Figure 4.** Map of (a) 500m PEL ice thickness DEM derived using mass conservation; (b) 1km PEL bed
DEM for the PEL sector; Profile A–A', B–B', C–C', D–D' and E–E' are overlain in (b); (c) 1km Bedmap2
bed elevation model (Fretwell et al., 2013), the red box indicates a location of a previously discovered
smooth-surface elongated and extensive feature interpreted as a potential subglacial lake (Jamieson
et al., 2016); (d) 1km Difference map between the PEL and Bedmap2 DEMs; (e) Subglacial water
pathway calculated with PEL (blue) and Bedmap2 (red) bed DEMs; (f) PEL hydropotential surface; and
(g) Bedmap2 hydropotential surface.




(a)

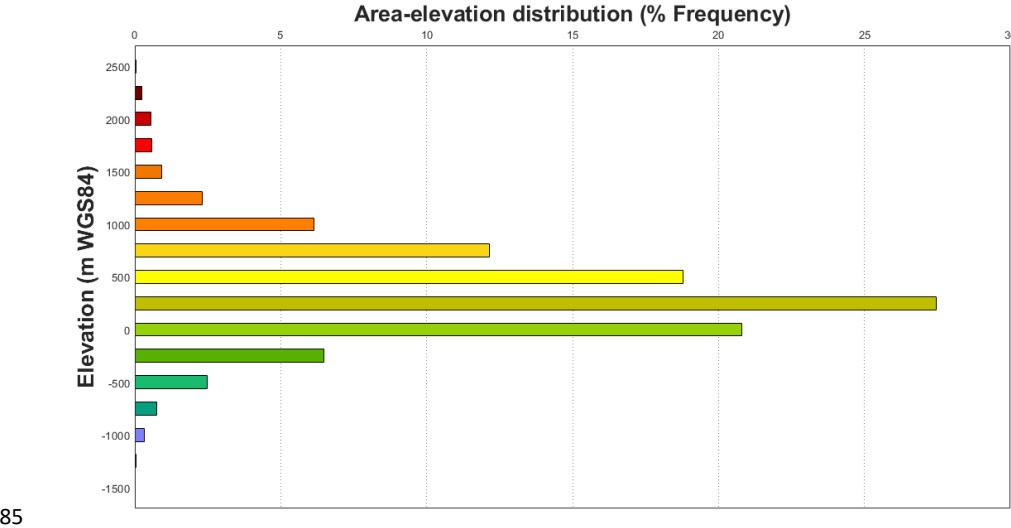


(b)

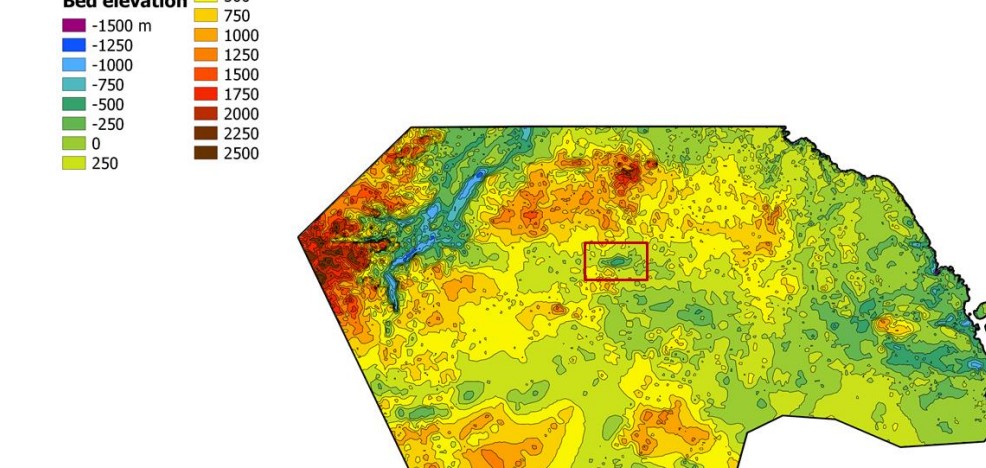


**Figure 5.** (a) Hypsometry (area-elevation distribution) derived from the PEL bed elevation model; and
(b) Bed elevation model determined for the PEL sector, East Antarctica, the red box indicates a location
of a previously discovered smooth-surface elongated and extensive feature interpreted as a potential
subglacial lake (Jamieson et al., 2016). The graph and map have the same elevation-related colour
scheme.



(a)

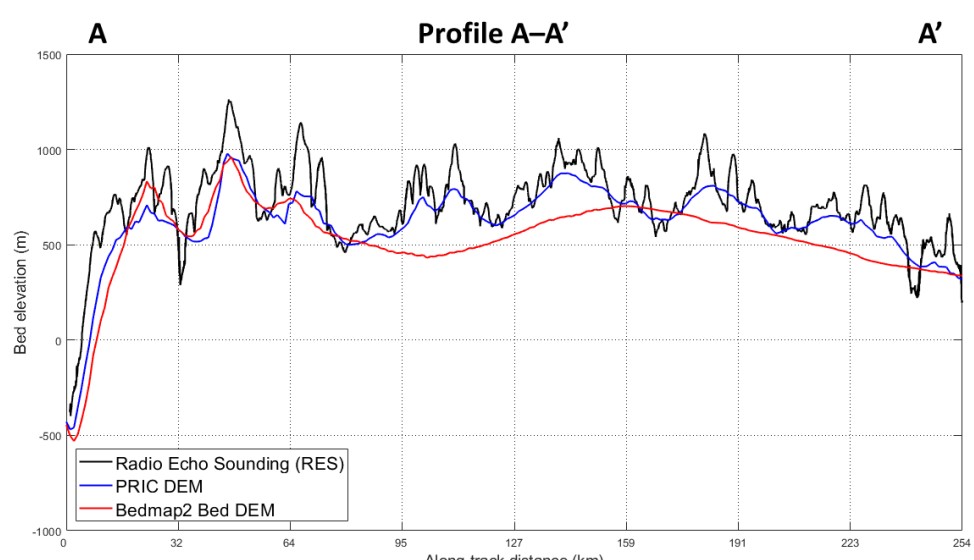

(b)

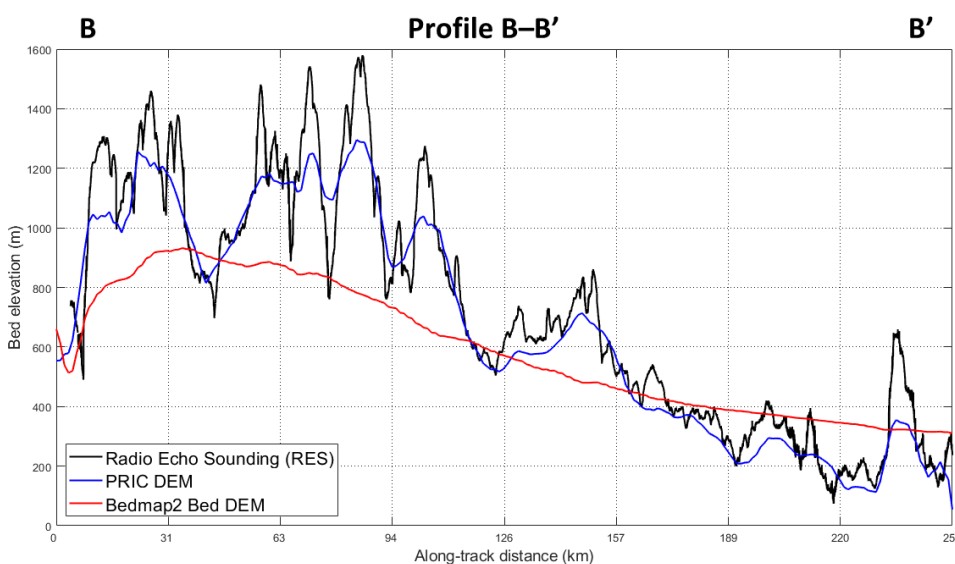




(c)

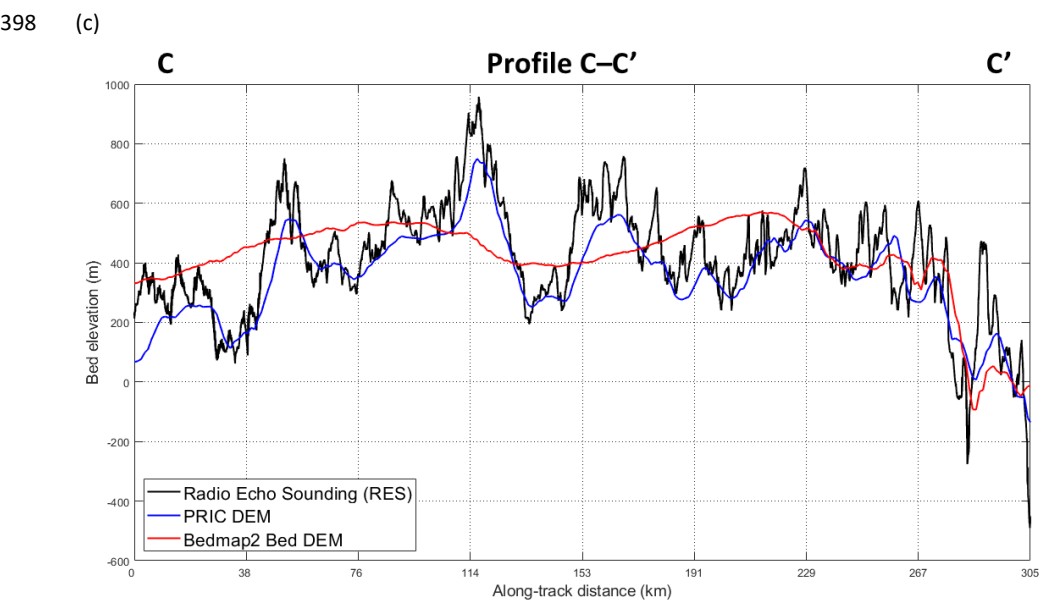

(d)

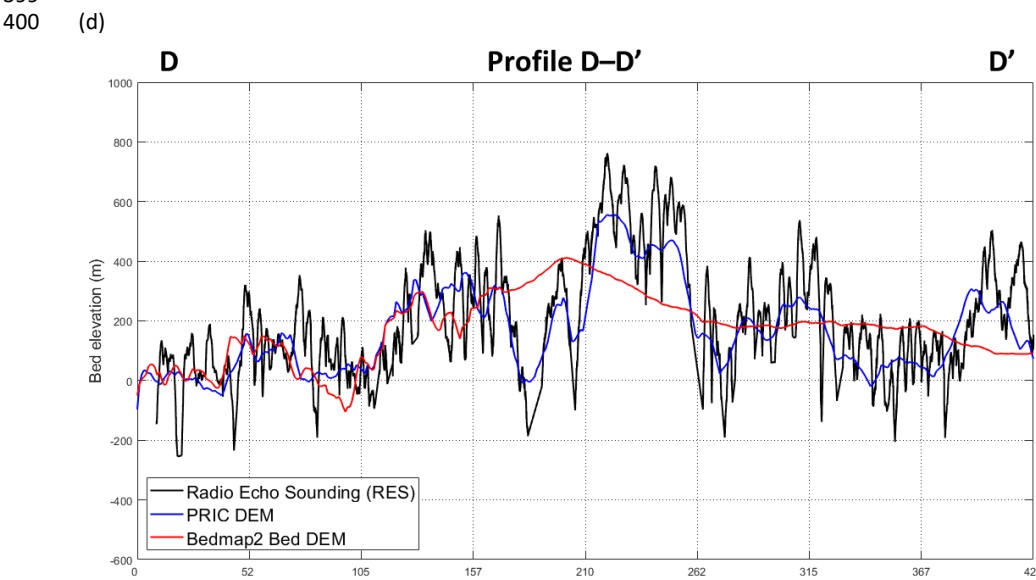


(e)

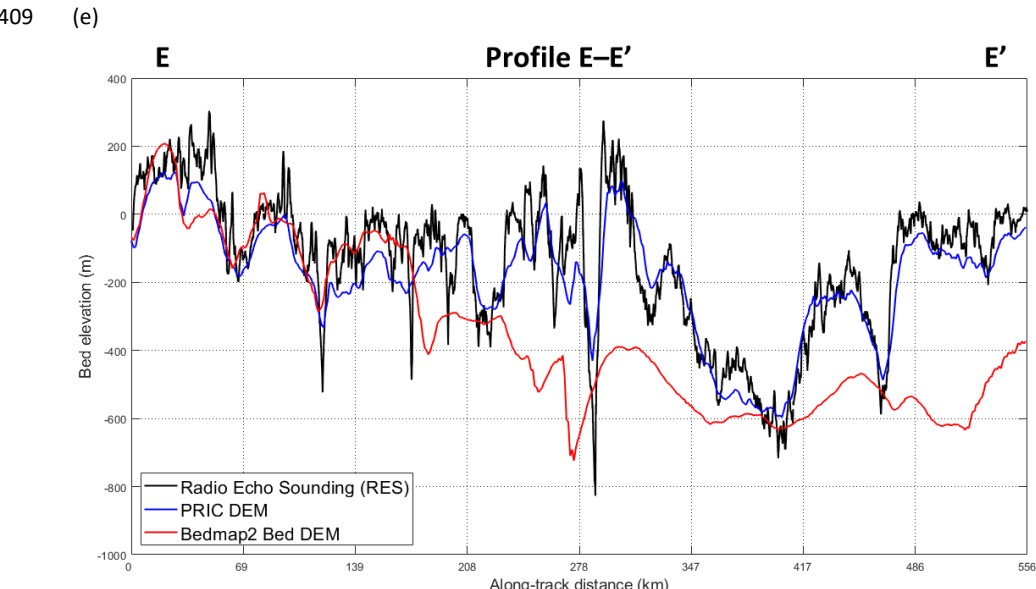

**Figure 6.** Bed elevations for RES transects (black), PRIC DEM (blue) and Bedmap2 (red) for (a) Profile
A–A', (b) Profile B–B', (c) Profile C–C', (d) Profile D–D' and (e) Profile E–E'.





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
