# Peer review of "Bed topography of Princess Elizabeth Land in East Antarctica"

_Earth System Science Data, 2020_

## Referee Comment (RC1) · Anonymous Referee #1 · 13 Jul 2020

This manuscript presents new and exciting radio echo sounding data from one of the most sparsely observed regions in Antarctica. Overall, this is a useful and unique data contribution to upcoming topographic compilations like BedMap3 and future versions of BedMachine Antarctica. There are, however, three major issues with the manuscript that would be valuable to address.

First, in the abstract and throughout the manuscript, the authors state that PEL is "the last remaining region in Antarctica to be surveyed by airborne radio-echo sounding techniques." This is a needless over-assertion which weakens the credibility of the authors and manuscript.

Second, the authors highlight the importance of collecting new data rather than using "inversion from poor resolution satellite gravity observations, and ice-flow modelling to

infer the subglacial landscape", but then use mass conservation, which is a ice-flow modelling inversion, to interpolate their data. This interpolated data is then compared to BedMap2, which is really a straw-man basis of comparison. The manuscript would me much stronger if it compared their presented topography to BedMachine Antarctica. This would show the actual value of the new data.

Finally, the authors use D8 routing for their hydrology analysis rather than more sophisticated routing algorithms or subglacial hydrology models. This needless open questions about whether their results are expressions of the topography or limitations of the overly simple routing.

---

## Referee Comment (RC2) · Robert Bingham (Referee) · 15 Jul 2020

This paper essentially provides to the community a new, detailed ice-thickness (and therefore subglacial topography) dataset for the last major data gap in Antarctica. Although the nuances of what constitutes a data gap can (and should!) be debated, this dataset certainly fills the last remaining "pole of ignorance" that was highlighted in Bedmap2 (Fretwell et al., 2013). As such, the product at the heart of this paper represents a highly anticipated and hugely valuable dataset; indeed, it epitomises what is to my mind the purpose of an ESSD paper, namely to fast-track dissemination of a much needed dataset to the wider community.

With regard to the paper itself, this is largely a straight-up presentation of the data to

the community, not couched in terms of addressing a major scientific hypothesis or question, and not stressing any major new "discoveries". In this context, I really don't have many criticisms, although I have annotated into the paper itself (as a supplement) a number of minor remarks for the authors to consider. One issue is that I don't think that nationalities/institutes/names of researchers involved in this or previous surveys need to appear so much in the main text of the manuscript – often affiliations and citations give this information anyway, and as international scientists working together I don't see the need to over-stress our nationalities or identities. I think there are various tweaks to be made to some of the figures – certainly it would be helpful to implement a scheme of being able to label locations on all maps, rather than the reader forever scrolling back to Figure 1 to find places.

Overall I enjoyed the paper, and of course the figures, not to mention the data, are the real gems here. I have a huge appreciation for the level of effort that has gone into acquiring these data in the first place. Congratulations to all concerned and all who have supported this stunning effort.

Please also note the supplement to this comment:
https://essd.copernicus.org/preprints/essd-2020-126/essd-2020-126-RC2-supplement.pdf
* * *
[Figure]

**Supplement:**

**Bed topography of Princess Elizabeth Land in East Antarctica 1**

Xiangbin Cui1, Hafeez Jeofry2,3, Jamin S Greenbaum4, Jingxue Guo1, Lin Li1, Laura E Lindzey5, Feras A Habbal6, Wei Wei4, Duncan A Young4, Neil Ross7, Mathieu Morlighem8, Lenneke M. 4 Jong9,10, Jason L Roberts9,10, Donald D Blankenship4, Sun Bo1 and Martin J. Siegert11 5

[revised manuscript text omitted]
                          | 500m bed            | Zenodo Data Repository | https://doi.org/10.5281/zenodo.3666    |  |
| DEM                                    | elevation DEM       | Cui et al. (2020)      | 088                                    |  |
| Ice thickness DEM                      | 500m ice            | Zenodo Data Repository | https://doi.org/10.5281/zenodo.3666    |  |
|                                        | thickness DEM       | Cui et al. (2020)      | 088                                    |  |
|                                        | Polar Research      |                        |                                        |  |
| Airborne ice                           | Institute of China  | Zenodo Data Repository | https://doi.org/10.5281/zenodo.3815    |  |
| thickness data                         | ice thickness data  | Cui et al., (2020)     | 064                                    |  |
|                                        | in CSV format       |                        |                                        |  |
| 1 km ice sheet                         | ERS-1 radar and     | National Snow and Ico  | https://nsidc.org/data/docs/daac/nsid  |  |
|                                        | ICESat laser        | Data Center (NSIDC)    |                                        |  |
|                                        | satellite altimetry |                        |                                        |  |
| Ice velocity map                       | MFaSUREs InSAR-     | National Snow and Ice  | https://doi:10.5067/MEASURES/CRYO      |  |
| of Central                             | hased ice velocity  | Data Center (NSIDC)    | SPHERE/nside-0484 001                  |  |
| Antarctica                             | based lee velocity  |                        |                                        |  |
| Ice sheet surface
satellite imagery | MODIS Mosaic of     |                        |                                        |  |
|                                        | Antarctica          | National Snow and Ice  | https://doi.org/10.7265/N5KP8037       |  |
|                                        | (2008 – 2009)       | Data Center (NSIDC)    | 11(1ps.//doi.org/10./203/10307803/     |  |
|                                        | (MOA2009)           |                        |                                        |  |
|                                        | RADARSAT (25m)      | Byrd Polar and Climate | https://research.bpcrc.osu.edu/rsl/rad |  |
|                                        | satellite imagery   | Research Center        | arsat/data/                            |  |

(a)

---

## Short Comment (SC1) · 15 Jul 2020

Many thanks for your thoughtful comments.

On comparison with Bedmachine Antarctica, this is an excellent point - thank you. We'll incorporate that assessment in the revision, but will retain the Bedmap2 comparison also.

On flow routing, again, we'll work on how to improve the hydrological model - but we'll also consider whether we need to retain this given the expansion to include Bedmachine comparison. We'll do the work suggested, but we're unsure at this stage on whether we'll include it or not.

One the mention of the survey completing the first order coverage of Antarctica, we

would prefer to keep this - because it has some significance. This really is the last big piece of the Antarctic bed jigsaw and we fell it deserves mention. We certainly don't claim that the survey is any better than any other - just the fact that bedcover in Antarctica has now been measured at least to a first degree with no major data gaps (there are always gaps - but smaller ones!).

Hope this indication of how we'll proceed is OK.

Thanks again,

Martin

---

## Author Comment (AC1) · 5 Sep 2020

We thank Robert Bingham for these supportive comments.

We will endeavour to minimise our statements around institutions and nationalities, as requested.

We will also adhere to all of the minor edits as recorded in the pdf version of the paper.

We don't feel there are many things to change - but we can certainly tighten up the paper based on these helpful remarks.

———————————————

---

## Author Comment (AC2) · 6 Sep 2020

We thank both referees for constructive and supportive reviews.

For referee 1, we will use Bedmachine Antarctica in addtion to Bedmap2 to highlight the improvements in our depiction of Antarctic bed topography. We don't see any major changes to the outcomes of the paper, as Bedmachine Antarctica does not use the new data reported here.

On hydrology, again, we will use an alternative flow router. The purpose of our examination of hydrology is merely to explain how calculations of water routing will be affected by the new topography. We don't see any major changes to the outcomes of the paper here either.

On statements about this being the final section of bed to be covered by RES data - we stand by that, and we feel it would be of interest generally to point this out.

For referee 2 (Rob Bingham), there are very few changes required. We will reduce mention of the various institutions, as recommended. We will also make minor edits suggested in a pdf version of the paper supplied by the referee.

In making these changes, we feel the paper will be tightened up - and improved - but not altered substantially.

Again, we really appreciate the time spent on reviewing our work.

Martin Siegert 6 Sept 2020

---

## Author Response (AR1)

**Grantham Institute**
**Climate Change and the Environment**
Imperial College London

South Kensington Campus
London SW7 2AZ
Tel: +44 20 7594 9666
Fax: +44 20 7594 9668

**Martin Siegert FRSE**
Co-Director and Professor of Geosciences m.siegert@imperial.ac.uk
www.imperial.ac.uk/climatechange

8th September 2020

To the editor, ESSD

**Bed topography of Princess Elizabeth Land in East Antarctica**

Many thanks for sending review of the above paper. We are pleased that both referees found the paper worthy of publication in ESSD, pending some changes. Please find uploaded a file containing details of how we have changed the above paper in accordance with referees' advice.

**Referee 1 (anon.)**

The main concern for referee 1, was that we hadn't used BedMachine Antarctica to compare the new results against. We have now included Bedmachine into the paper, as requested.

In the original paper we provided an assessment of basal hydrology. The referee recommended we chose an alternative algorithm to calculate the flow of water. To simplify and focus the paper – so that it simply presents the bed data – we have now taken out the basal hydrology component (including subglacial lakes).

On statements about this being the final section of bed to be covered by RES data - we stand by that, and we feel it would be of interest generally to point this out. However, we recognise that we have made too many references to national and organisational contributions and have removed much of these.

**Referee 2 – Robert Bingham**

For referee 2, there are very few changes required. We have reduced mention of the various institutions, as recommended. We have also made minor edits suggested in a pdf version of the paper supplied by the referee.

We hope that the paper is now ready for publication in ESSD. Do let us know if further modifications are necessary.

Yours sincerely,

Martin Siegert

[revised manuscript text omitted]
 between ICECAP2 bed DEM and both Bedmap2 and BedMachine bed DEMs has been identified and measured, and across the Wilhelm II Land toward the margin and near to the SPRI-60 subglacial lake between ICECAP2 and Bedmap2 bed DEMs. The ICECAP2 DEM completes the first-order data coverage of subglacial Antarctica – a feat spanning around 70 years of international collaboration.

**Acknowledgements**

This paper is a contribution of the ICECAP2 consortium (International Collaborative Exploration of Central East Antarctica through Airborne geophysical Profiling) led by SB, JLR, DDB and MJS. The research was supported by the Chinese Polar Environmental Comprehensive Investigation and Assessment Programs (CHINARE-02-02), the National Natural Science Foundation of China (41941006) and the National Key R&D Program of China (2019YFC1509102). MJS acknowledges support from the British Council's Global Innovation Initiative between the UK, USA, China and India. We thank the volunteers at QGIS for open-source software used to draw many of the figures in this paper. DDB, JG and DY acknowledge the G. Unger Vetlesen Foundation, and US National Science Foundation grants PLR-1543452 and PLR- 1443690. JR acknowledges the Australian Antarctic Division, which provided funding and logistical support (AAS 4346 and 4511). This work was also supported by the Australian Government's Cooperative Research Centres Programme through the Antarctic Climate & Ecosystems Cooperative Research Centre and under the Australian Research Council's Special Research Initiative for Antarctic Gateway Partnership (Project ID SR140300001). This is UTIG contribution ####.

**Competing Interests**

The authors report no competing interests for this paper.

**Author contributions**

This paper was research and written by the ICECAP2 (partnership, in which all authors are members. Specific responsibilities are as follows. XB, JSG, JG, LL, LEL, FH, WW, LJ and JRL undertook fieldwork and data acquisition. JSG and DAY undertook data processing. MM and HJ undertook data

264 interpolation. All authors comments and edited drafts of this paper. The paper was written by MJS
265 and HJ.

**Table 1:** Data files and locations.

| Products | Files | Location | DOI/URL |
|---|---|---|---|
| Bed elevation DEM | 500 m bed elevation DEM | Zenodo Data Repository Cui et al. (2020) | https://doi.org/10.5281/zenodo.3666 088 |
| Ice thickness DEM | 500 m ice thickness DEM | Zenodo Data Repository Cui et al. (2020) | https://doi.org/10.5281/zenodo.3666 088 |
| Airborne ice thickness data | Polar Research Institute of China ice thickness data in CSV format | Zenodo Data Repository Cui et al., (2020) | https://doi.org/10.5281/zenodo.3815 064 |
| 1 km ice sheet surface DEM | ERS-1 radar and ICESat laser satellite altimetry | National Snow and Ice Data Center (NSIDC) | https://nsidc.org/data/docs/daac/nsid c0422_antarctic_1km_dem/ |
| Ice velocity map of Central Antarctica | MEaSUREs InSAR-based ice velocity | National Snow and Ice Data Center (NSIDC) | https://doi:10.5067/MEASURES/CRYO SPHERE/nsidc-0484.001 |
| Ice sheet surface satellite imagery | MODIS Mosaic of Antarctica (2008 – 2009) (MOA2009) | National Snow and Ice Data Center (NSIDC) | https://doi.org/10.7265/N5KP8037 |
| | RADARSAT (25m) satellite imagery | Byrd Polar and Climate Research Center | https://research.bpcrc.osu.edu/rsl/rad arsat/data/ |

(a)

[Figure]

(b)

(c)

[Figure]

**Figure 1.** Map of (a) ice flow velocity version 2 (Rignot et al., 2017b); (b) MODIS Mosaic of Antarctica
2008–2009 satellite image (Haran et al., 2014). The black line denotes the grid boundary for ICECAP2
bed elevation model White box indicates a location of a previously discovered smooth-surface
elongated and extensive feature interpreted as a potential subglacial lake (Jamieson et al., 2016); and
(c) the Aerogeophysical flight lines surveyed by PRIC in four seasons which are 2015/16 (orange),
2016/17 (green), 2017/18 (red) and 2018/19 (blue) across the PEL sector; the inset denotes location
of the study region in East Antarctica. Figures 1b and 1c are overlain by MODIS Mosaic of Antarctica
2008–2009 (Haran et al., 2014). The Differential Interferometry Synthetic Aperture Radar (DInSAR)
grounding line (yellow line) are also shown (Rignot et al., 2017a).

(a)

[Figure]

(b)

[Figure]

(c)

[Figure]

**Figure 2.** (a) Snow Eagle 601 airplane operated by the Polar Research Institute of China for the Chinese
National Antarctic Research Expedition (CHINARE) program; (b) The interior image of the airplane
showing the airborne radio-echo sounder equipment; and (c) Two-dimensional radio-echo sounding
radargram collected in 2017/18 revealing the quality of internal layers, bed topography and subglacial
lake water.

[Figure]

**Figure 3.** Map shows interpolation techniques used to infer ice thickness DEM across PEL, reference
Elevation Model of Antarctica, International Bathymetric Chart of the Southern Ocean (REMA IBCSO,
green), mass conservation (brown), interpolation (yellow) and streamline diffusion (blue).

(a)

[Figure]

[Figure]

(b)

[Figure]

[Figure]

(c)

[Figure]

(d)

[Figure]

(e)

[Figure]

[Figure]

[Figure]

(f)

[revised manuscript text omitted]